# Hybrid ZrO₂/Cr₂O₃ Epoxy Nanocomposites as Organic Coatings for Steel

**Ayman M. Atta [1],\*** , **Mona A. Ahmed [2]**, **Ashraf M. El-Saeed [2]**, **Ossama M. Abo-Elenien [2]** and **Maher A. El-Sockary [2]**

[1] Department of Chemistry, College of Science, King Saud University, Riyadh 11451, Saudi Arabia
[2] Egyptian Petroleum Research Institute (EPRI), Nasr City 11727, Cairo, Egypt;
mona_chemist17@yahoo.com (M.A.A.); ashrfelsaied@yahoo.com (A.M.E.-S.);
drossamaa@hotmail.com (O.M.A.-E.); delsockary@gmail.com (M.A.E.-S.)
\* Correspondence: aatta@ksu.edu.sa

**Abstract:** Mixed ZrO₂ and Cr₂O₃ nanoparticles (NPs) were prepared using a liquid phase chemical technique and applied as reinforced filler for epoxy coatings with different weight ratios ranged from 0.5 to 2.5 wt.% to protect carbon steel from corrosion. The ZrO₂/Cr₂O₃ NPs were used to catalyze the curing of the epoxy composite films to modify their mechanical and thermal characteristics on the steel surface. The crystalline structure, particle sizes, and surface morphologies of the prepared ZrO₂ and Cr₂O₃ NPs were characterized to investigate their chemical composition and dispersion. The thermal stability of epoxy ZrO₂/Cr₂O₃ coating films was investigated by thermogravimetric analysis (TGA), and the mechanical properties of the cured epoxy films were also studied. The dispersion of the Cr₂O₃/ZrO₂ NPs into the epoxy matrix was investigated by scanning electron microscope (SEM), dynamic mechanical analysis (DMA) and TGA analyses. The results of salt spray test, used to investigate the anticorrosion performance of the epoxy coatings) were combined with thermal characteristics to confirm that the addition of Cr₂O₃/ZrO₂ NPs significantly improved the corrosion resistance and the thermal stability of epoxy coating. The mechanical properties, adhesion, hardness, impact strength, flexibility and abrasion resistance were also improved with the addition of ZrO₂/Cr₂O₃ NPs filler content.

**Keywords:** ZrO₂ nanoparticles; Cr₂O₃ nanoparticles; epoxy nanocomposite; thermal stability; corrosion resistance

## 1. Introduction

Epoxy resins have interesting mechanical and barrier properties to protect several substrates from corrosion in aggressive environment beside they have interesting adhesive properties due to their high modulus, high failure strength, and low creep mechanical characteristics [1–3]. The epoxy coating performances were based on their good adhesion and physical barrier between epoxy and the substrate surface against several corrosive environments [2,3]. Nevertheless, the microstructure of cured epoxy resin has an undesirable high brittleness and poor resistance to crack initiation and growth due to their fast curing with different hardeners based on polyamines and polyamides as curing agent [1–3]. The formation of micro-cracks and holes due to fast curing of heterogeneous epoxy networks facilitates the penetration of corrosive electrolytes containing oxygen, water, and ions from coat to substrate surface when epoxy coatings exposed against aggressive environmental to a long time [2]. Moreover, the penetration of the corrosive ions into the primer epoxy coatings reduces the adhesion of cured epoxy with substrate surfaces [2,3]. It is reported that the chemical treatment of the surface substrate beside reinforcement of the epoxy with nano-fillers can improve the adhesion of epoxy primers with

several metallic substrates [4–15]. Sababi et. al. [5] reported that the adhesion and wet durability of the fusion bonded epoxy improved with carbon steel substrate upon exposure to a sodium chloride electrolyte when they treated with zirconium-based materials. These materials were used to inhibit the formation of large water aggregates and to prevent the epoxy disbonding at substrate/epoxy interface [5]. Ramezanzadeh et al. [6] reported the pre-treatment hot dip galvanized specimens of with Cr (III) and Cr (VI) was used to improve the anticorrosion performance and adhesion properties of the epoxy nanocomposites embedded with ZnO NPs [6]. Asemani et al. [7] and Hosseini et al. [8] revealed that the treatment of the carbon mild steel surface with hexafluorozirconic acid had better performance of epoxy coatings in a salt spray test and electrochemical measurements against corrosive media. It was also reported that the Zr-based conversion layer improved adhesion of the epoxy paints with the steel surfaces [9,10]. The zirconium-based conversion layer is responsible on the formation of zirconium oxide–hydroxide on the substrate surfaces to increase the adhesion of the substrate surfaces with epoxy organic coating [11]. The treatment of zirconia NPs with aminopropyltrimethoxy silane (APS) to embed in the epoxy networks was used to produce improve epoxy coating toughness [12]. Moreover, zirconia NPs were used to enhance the barrier properties, anticorrosion performance, and Ohmic resistance of epoxy coatings as compared with neat-epoxy coating, via [13,14]. The mechanical properties and surface morphologies of $ZrO_2$ epoxy nanocomposites have been investigated [15]. It was found that, the toughness of an epoxy resin was improved by increasing the content of the $ZrO_2$ NPs due to their bonding with epoxy matrix. So, in this perspective epoxy based nanocomposite coating containing various combinations of zirconia and chromium nanoparticles were prepared to enhance the structure and thermal stability of epoxy nanocomposite.

The addition of Cr nanoparticles into epoxy resin can further used to improve its resistance toward corrosion due to greatly improvement the corrosion resistance due to the "self-healing" effect [16,17]. The main composition of trivalent chromium process (TCP) coating is a mixed oxide/oxide of zirconium-chromium (III) appeared to provide some active protection [18]. It was referred for the localized transient formation of mobile Cr(VI) species after immersion in aggressive solutions [19,20]. In this work $Cr_2O_3$ and $ZrO_2$ nanoparticles were synthesized separately, characterized and added at different ratios to epoxy resins. The epoxy composites were cured and applied on the surface metal specimens protects surface metal from corrosion. The flexibility and rigidity of the epoxy coating composites, based on their mechanical characteristics and thermal stabilities, were investigated by dynamic mechanical analyzer (DMA), and thermogravimetric analysis (TGA). Finally, the corrosion properties of the epoxy coating composites were measured by salt spray and their mechanical properties as impact pull off and hardness tests were evaluated.

## 2. Experimental

### 2.1. Materials

All the materials used in this work were purchased from Sigma Aldrich Chemicals Co. and used without further purification. Zirconium oxychloride octahydrate ($ZrCl_2O \cdot 8H_2O$), chromium sulfate $Cr_2(SO_4)_3$, ammonium hydroxide $NH_4OH$ (ammonia solution 25%), and deionized water were used to prepare chromium oxide ($Cr_2O_3$) and zirconium oxide ($ZrO_2$) nanoparticles. Commercial two components, solvent-less epoxy resin Epikote resin 828 (Hexion, Olana, Italy) and polyamine hardener were used as the organic coating matrix. The typical characteristics of the epoxy coating are as follows: epoxy equivalent weight (190–200 g/eq), viscosity at 25 °C (13.0–15.0 Pa·s), density at 25 °C (1.17 kg/L) and mixing ratio by weight of epoxy resin to hardener is 4:1 (wt.%).

## 2.2. Methods and Techniques

### 2.2.1. Synthesis of Zirconium Oxide Nanoparticles (ZrO$_2$ NPs)

ZrCl$_2$O$_8$H$_2$O (0.1 mol) was added dropwise to NH$_4$OH (NH$_3$ 25 wt.%; 250 mL) and the pH was adjusted to 10.5 under vigorous stirring for 2 h. The solid particles were separated from the suspension by ultracentrifuge at 5000 rpm for 20 min and washed with ethanol three times. The obtained ZrO$_2$ nanoparticles was kept at 70 °C for 12 h in a vacuum oven, then calcined at 400 °C for 2 h.

### 2.2.2. Synthesis of Chromium Oxide Nanoparticles (Cr$_2$O$_3$NPs)

NH$_4$OH solution (100 mL) was added dropwise to Cr$_2$(SO$_4$)$_3$, 250 mL of 0.1 M solution, with vigorous stirring and adjusted the pH 10. The obtained precipitates were filtered and then washed with distilled water. The precipitates were dried in an oven at 70 °C for 24 h and calcined at 600 °C in a muffle furnace for 5 h.

### 2.2.3. ZrO$_2$/Cr$_2$O$_3$ Epoxy Nanocomposite Coating Films

ZrO$_2$/Cr$_2$O$_3$ epoxy coating films were prepared by dispersing different loadings of ZrO$_2$/Cr$_2$O$_3$ NPs (equal weight ratio) ranged from 0.5 to 2.5 wt.% related to the total weight of both epoxy and polyamine hardener. The NPs were dispersed in in xylene solvent by continuously sonication, using sonicator (model Sonics & Materials, VCX-750, Newtown, CT, USA) utilized a frequency of 20 kHz, equipped with a 13 mm diameter titanium probe, for 25 min. The dispersed ZrO$_2$/Cr$_2$O$_3$ NPs were mixed with epoxy base component for 20 min by stirring. The polyamine hardener was mixed with ZrO$_2$/Cr$_2$O$_3$ epoxy solution at mixing ratio (1:4 wt.%) under continuous stirring. The mixed solution was sprayed on the cleaned steel panels to form a uniform dry film thickness (DT) of 100 μm for epoxy coatings. The same quantity of mixed solutions sprayed on the panels at the same distance from the spray nozzle to panels were applied to control the cured film thickness. The thermal stability, mechanical properties and anticorrosion properties of the cured epoxy nanocomposites as well as epoxy without filler (blank) were evaluated after 7 days of curing at room temperature.

## 2.3. Characterization Study of the Prepared ZrO$_2$ NPs and Cr$_2$O$_3$NPs

Crystal structure of the prepared ZrO$_2$ NPs and Cr$_2$O$_3$ NPs was identified by X-ray diffraction (XRD) patterns using (a X'Pert, Philips, Amsterdam, The Netherlands) employing Cu-Kα radiation at 50 kV and 200 mA. Transmission electron microscopy (TEM; JEM2100 LaB6, Tokyo, Japan) was used to investigate the morphology of ZrO$_2$ and Cr$_2$O$_3$ NPs. The particle size, and polydispersity index (PDI), of ZrO$_2$/Cr$_2$O$_3$ NPs were determined using dynamic light scattering (DLS) (Malvern Instrument Ltd., London, UK). The surfaces morphologies of ZrO$_2$/Cr$_2$O$_3$ epoxy nanocomposite coated films were measured using a scanning electron microscope (SEM, model Quanta 250 FEG, FEI, Eindhoven, The Netherlands). The thermal stability of the blank epoxy and the cured ZrO$_2$/Cr$_2$O$_3$ coated films were obtained using thermogravimetric analysis (TGA; NETZSCH STA 449 C instrument, New Castle, DE, USA) with a temperature rate of 10 °C/min, under dynamic flow of nitrogen 20 mL/min. The storage modulus (È), and damping factor (tanδ) were measured for the prepared epoxy nanocomposites using dynamic mechanical analyzer (DMA; Triton Technology-TTDMA, Mansfield, MA, USA). Three points bending mode with frequency of 1 Hz was applied. The dimension of specimens are 25 mm length, 10 mm width and 3 mm thickness. The samples were exposed to 100 °C with heating rate of 5 °C/min.

## 2.4. Mechanical and Corrosion Resistances Property of ZrO$_2$/Cr$_2$O$_3$ Coating Films

Posi Test AT-A Automatic Adhesion Tester was used to evaluate the adhesion pull-off strength of cured epoxy coating according to American Society for Testing and Materials (ASTM D 4541-17) [21]. Flexibility bend test, impact resistance and scratch hardness resistance was evaluated using ASTM

D522 / D522M-17 [22], ASTM D2794-93(2019) [23] and ASTM D 3363-20 [24]. The abrasion resistance of coating films was evaluated according to ASTM 4060-19 [25].

The corrosion behavior of the coating films was investigated according to ASTM B 117-19 [26]. The coated panels were exposed to a 5 wt.% NaCl salt spray (fog) solution at 37 °C for 720 h in a cabinet manufactured by CW specialist equipment Ltd., 20 Model SF/450, (London, UK). The corrosion resistance was determined by the rate of failure at scribe (ASTM D-1654 [27]) of the 6 steel panels.

## 3. Results and Discussion

### 3.1. Characterization of the Prepared $ZrO_2$ and $Cr_2O_3$ NPs

The crystalline lattice structure of $ZrO_2$ and $Cr_2O_3$ NPs was confirmed from XRD data represented in Figure 1a,b, respectively. Figure 1a. shows the formation of two crystalline phases of $ZrO_2$ based on cubic (red square) and tetragonal lattice structures (blue square) correspond well to (JCPDS-37-14844), and JCPDS 01-(JCPDS-17-0923), respectively. By comparing the intensities of the two crystalline phases, it can be found that the predominant good crystalline cubic phase obtained with small fraction in the tetragonal phase. The calculated ratio for the cubic to tetragonal phases is 8:1. The broadening of $ZrO_2$ XRD peak (Figure 1a) elucidates the formation of fine nanoparticles. The diffraction characteristic peaks of the cubic crystalline structure at different 2θ and their planes were clarified in Figure 1a. The diffraction patterns of the cubic phase of $ZrO_2$ nanoparticles coincide with the standard data of (JCPDS-37-1484) with lattice parameters a = 5.3125 Å, b = 5.2125 Å and c = 5.1477 Å [28,29]. The diffraction patterns of the tetragonal phase are in good agreement with the standard data of (JCPDS-17-0923) For $ZrO_2$ nanoparticles. It was previously reported that the cubic and tetragonal phases are unstable at ambient temperature but if the particle size is less than 30 nm, the tetragonal phase can be formed at room temperature [30]. This work elucidates that $ZrO_2$ nanocrystals were produced with anisotropic shapes and various crystal structures that also obtained when hydrothermal process was used [31]. Figure 1b shows the formation of crystalline $Cr_2O_3$ with the formation of rhombohedral phase (JCPDS no. 38-1479 with a = 4.95876 Å, b = 13.594 Å and space group R3-c) [32]. The diffraction characteristic peaks of the rhombohedral phase at different 2θ and their planes were clarified in Figure 1b [24]. The crystalline sizes of the $ZrO_2$ and $Cr_2O_3$ NPs were calculated using the Scherrer formula, $D = k\lambda/\beta\cos\theta$; where λ is the X-ray wavelength (1.5406 Å), K is a constant (0.9), θ is the Bragg diffraction angle, and β is the pure diffraction broadening peak located at half-height. The most prominent peaks at (111) and (104) for $ZrO_2$ and $Cr_2O_3$ NPs (Figure 1a,b, respectively) were used to calculate the average crystal sizes, that are found to be around 18.9 and 12.4 nm, respectively.

The morphologies of the $ZrO_2$ and $Cr_2O_3$ NPs were evaluated from TEM micrograph as that represented in Figure 2a,b, respectively. Figure 2a shows an overall view of Zirconium oxide nanoparticles, revealing a large quantity of nanoparticles with small size distribution. The diameters of the particles materials are almost uniform around 25.4 nm. The transmission electron microscope provides as in Figure 2b that the average crystalline size calculated is 20.3 nm which is in close agreement with the XRD results of $ZrO_2$ and $Cr_2O_3$ NPs. These data elucidate that the proposed synthesis method in this work is an easy method to prepare $ZrO_2$ and $Cr_2O_3$ NPs. The morphology of the zirconium dioxide (Figure 2a) consisted mostly of cubic and tetragonal. However, it seems that even the $Cr_2O_3$ NPs (Figure 2b) are aggregates of smaller, 40–60 nm, than $ZrO_2$ (Figure 2a). It is clear that agglomeration takes place among $ZrO_2$ as a result of nanoparticle interaction more than that obtained among $Cr_2O_3$ NPs.

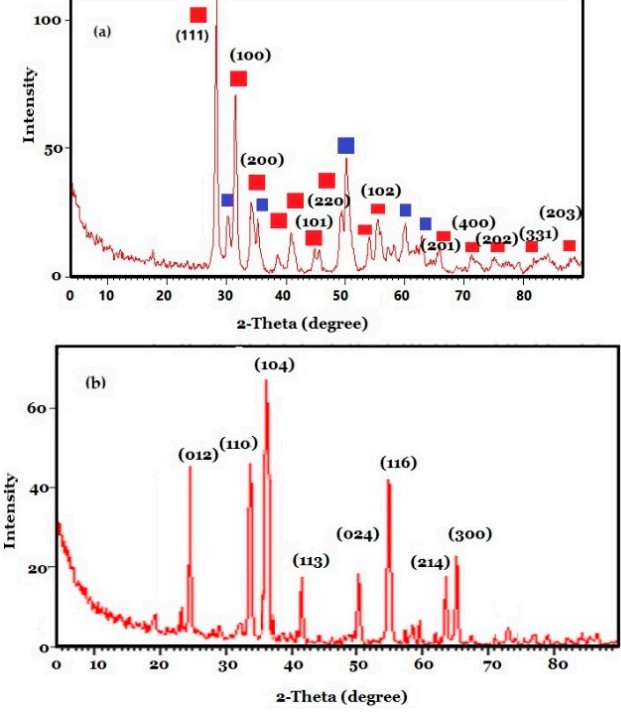

**Figure 1.** X-ray diffraction patterns of synthesized (**a**) $ZrO_2$ and (**b**) $Cr_2O_3$ NPs.

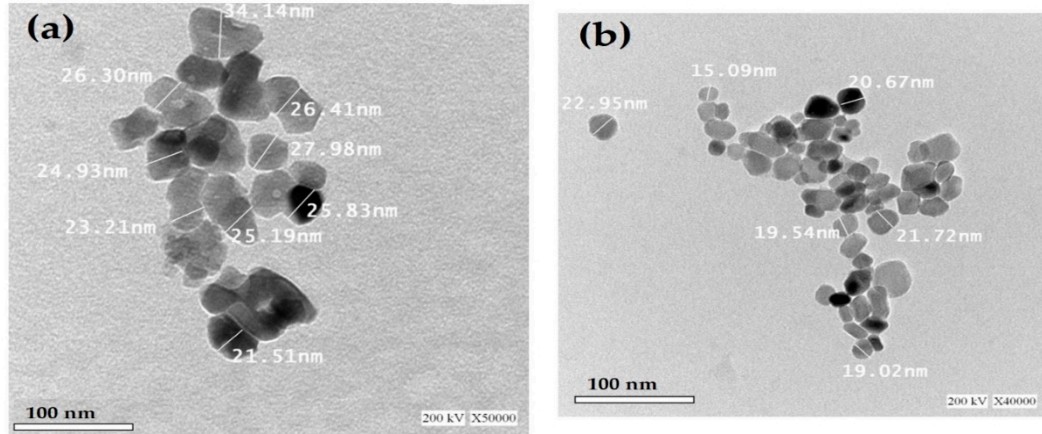

**Figure 2.** TEM for (**a**) $ZrO_2$ NP and (**b**) $Cr_2O_3$ nanoparticles (NPs) structure.

The particle sizes and polydispersity index (PDI) of $ZrO_2$ and $Cr_2O_3$ NPs in water were evaluated from DLS measurements as summarized in Figure 3a,b. The particle sizes were measured from the average hydrodynamic diameters that elucidate that the dimeters of $ZrO_2$ and $Cr_2O_3$ NPs are 375.14 and 120.30 nm, respectively. Moreover, the PDI data of $ZrO_2$ and $Cr_2O_3$ NPs are 0.532 and 0.320, respectively to confirm that the lower dispersion of $ZrO_2$ NPs than $Cr_2O3$ NPs in water. These data confirm that the $Cr_2O_3$ NPs form hydration layer at their surfaces more than the $ZrO_2$ NPs [32].

*3.2. Surface Morphology and Thermal Characteristics of $ZrO_2/Cr_2O_3$ Epoxy Nanocomposite Coating*

It was previously reported that the addition of zirconia to silica or clay nanoparticles leads to disturb curing procedure and decrease epoxy network crosslinking density and increase the barrier properties of the nanocomposites [33]. Moreover, the mechanical performances of an epoxy-based adhesive have been improved by the addition of zirconia NPs [34]. Figure 4a–d shows SEM micrographs prepared from the fractured surface of epoxy coating contains different weight % of $ZrO_2/Cr_2O_3$ ranged from 0.5 to 2.5 wt.%. It was noticed that $ZrO_2/Cr_2O_3$ NPs appeared as white powder embedded in the

epoxy matrix have uniform and dispersed distribution. Their diameters were determined as 50 nm without agglomeration.

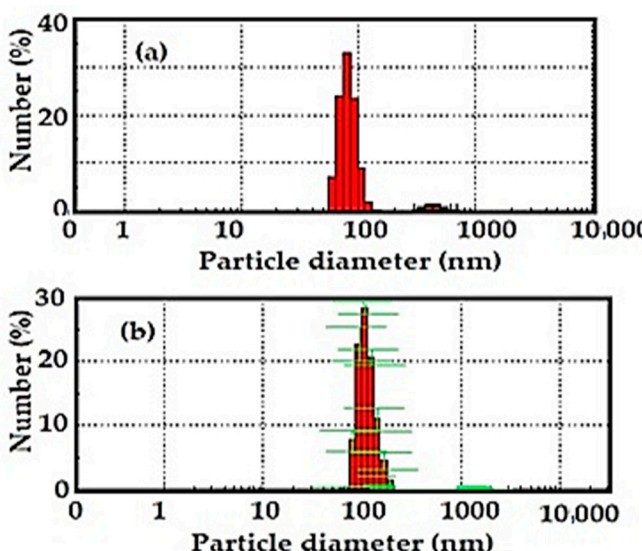

**Figure 3.** DLS data of (**a**) ZrO$_2$ NPs and (**b**) Cr$_2$O$_3$ NPs in 1 mM KCl aqueous solution at 25 °C.

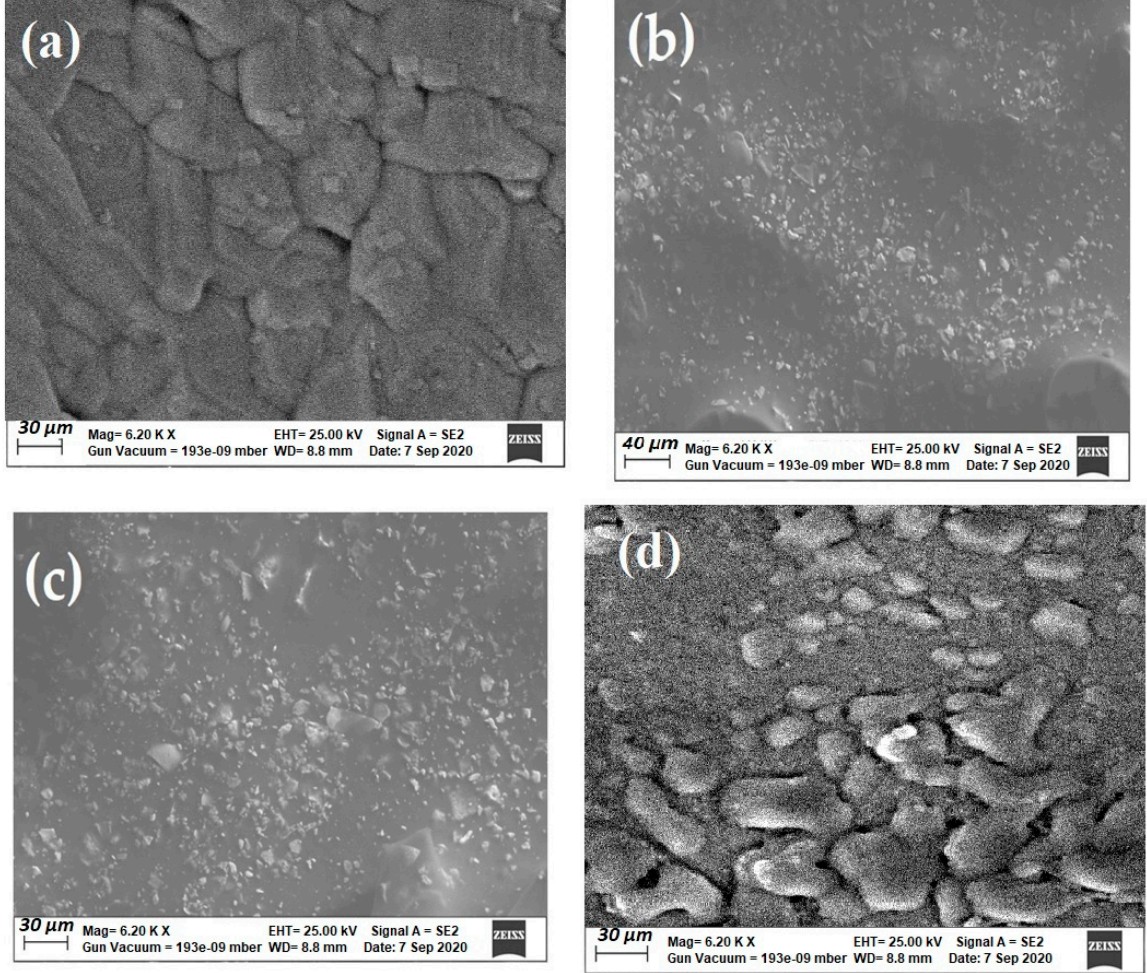

**Figure 4.** SEM for ZrO$_2$/Cr$_2$O$_3$ Epoxy nanocomposite coating mixed with different weight ratios (**a**) 0.5, (**b**) 1.0, (**c**) 1.5, and (**d**) 2.5 wt.%.

　　　TGA thermograms performed to obtain information on thermal stability of the epoxy nanocomposite coatings were summarized in Figure 5. The obtained results show that the increasing of $ZrO_2/Cr_2O_3$ NPs concentration from 0.5 to 2.5 wt.% increases the thermal stability of the epoxy. It is obtained from the peak of loss factor curves temperature that increased with the addition of $ZrO_2/Cr_2O_3$ from 235 °C to 285 °C at 10% weight loss. This increasing is due to the capability of $ZrO_2/Cr_2O_3$ NPs to catalyze the curing reaction of epoxy networks and its role as a thermal stabilizer [35]. The amount of char yields or residues at 650 °C was increased from 6.15 to 12.50 wt.%. This is due to, the use of NPs can lead to the formation of a barrier which can prevent the evolution of volatiles during the degradation and thus increases the amount of char that was produced [36].

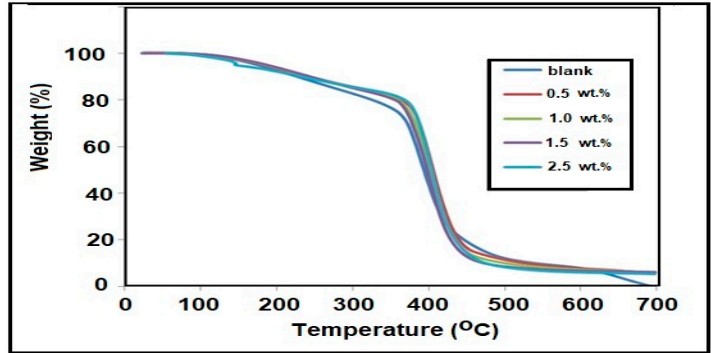

**Figure 5.** Thermogravimetric curves of blank epoxy and $ZrO_2/Cr_2O_3$ epoxy nanocomposites coating films containing different loading level of $ZrO_2/Cr_2O_3$ NPs.

　　　DMA measurements of the cured epoxy in the presence of different weight ratios of $ZrO_2/Cr_2O_3$ NPs were represented in Figure 6. The modulus of bending for the cured net epoxy resin, 0.5, 1, 1.5, and 2.5 wt.% of epoxy nanocomposites are 0.14, 0.19, 0.27, 0.37, and 0.54 GPa. These mean that the increasing of $ZrO_2/Cr_2O_3$ NPs contents into the epoxy composites enhances their modulus of epoxy as compared with neat epoxy. This was referred to the increasing of the epoxy crosslinking degree in addition to the good interfacial force between hybrid NPs and epoxy matrix [37]. It was also noticed that the loading of $ZrO_2/Cr_2O_3$ NPs up to 1.5 wt.% enhances the modulus of the epoxy nanocomposite to about 60%. The high loading of $ZrO_2/Cr_2O_3$ NPs up to 2.5 wt.% (Figure 6) decreases the modulus of bending for the epoxy nanocomposites due to agglomeration of $ZrO_2/Cr_2O_3$ NPs [38].

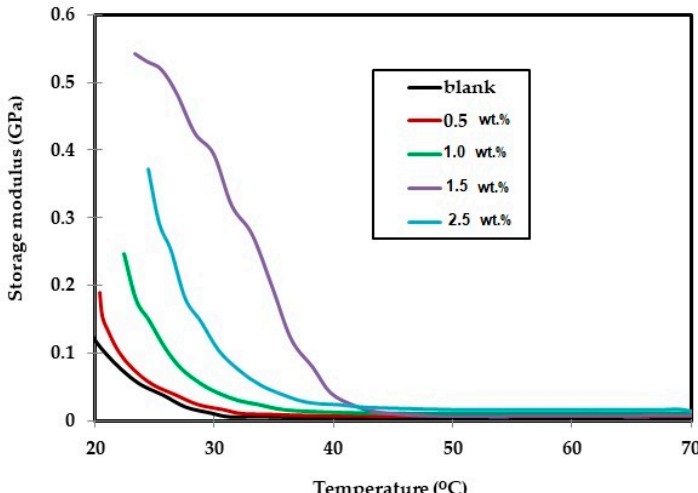

**Figure 6.** Effect of NPs loading on modulus of epoxy nanocomposites.

　　　The effects of the surface modification and loading percent of $ZrO_2/Cr_2O_3$ NPs on tan δ of epoxy composites were also discussed (as shown in Table 1, and Figure 7). The results confirm that the

incorporation of $ZrO_2/Cr_2O_3$ NPs into epoxy networks increases the dissipated energy (tanδ) and consequently the main mechanical relaxation is enhanced. Moreover, the $T_g$ values of the epoxy nanocomposites increase with increasing the loading of $ZrO_2/Cr_2O_3$ NPs to increase the rigidity of the epoxy composites. This can be attributed to the interfacial interaction between hybrid $ZrO_2/Cr_2O_3$ NPs and epoxy during the curing process. Data represented in Table 1 record higher $T_g$ values for 1.5% and 2.5% $ZrO_2/Cr_2O_3$ NPs epoxy nanocomposites as 30.3 and 35.6 °C, respectively as compared with that neat epoxy. The increasing e in $T_g$ was attributed to the ability of the $ZrO_2/Cr_2O_3$ NPs filler to hinder the thermal motion of the epoxy hosting matrix chains [13].

**Table 1.** $T_g$ results for $ZrO_2/Cr_2O_3$ epoxy nanocomposite coating films.

| Coating Design | NPs Weight % (wt.%) | $T_g$ (°C) |
|---|---|---|
| Blank epoxy (E) | 0 | 23.8 ± 0.2 |
| Epoxy/ZrO₂/Cr₂O₃ NPs | 0.5 | 24.6 ± 0.1 |
| | 1.0 | 25.9 ± 0.3 |
| | 1.5 | 30.3 ± 0.1 |
| | 2.5 | 35.6 ± 0.2 |

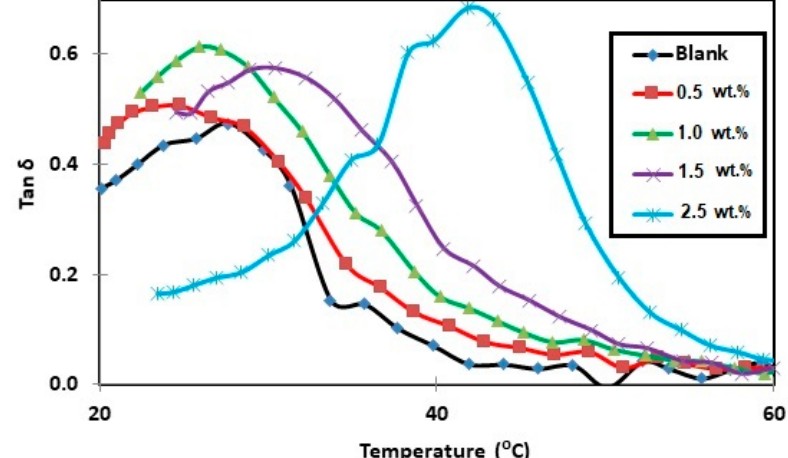

**Figure 7.** Effect of NPs loading on loss tangent (tanδ) of epoxy nanocomposites.

*3.3. Mechanical Properties and Anticorrosion Properties of $ZrO_2/Cr_2O_3$ Epoxy Nanocomposite Coating on the Steel Substrate*

The effect of $ZrO_2/Cr_2O_3$ NPs on the mechanical properties and adhesion strength of epoxy coating on the steel surfaces was summarized in Table 2. It was observed that the scratch hardness increases steadily with increasing $ZrO_2/Cr_2O_3$ NPs content at the loading level from 0.5 to 2.5 wt.%. However, the increasing the $ZrO_2/Cr_2O_3$ NPs filler content more than 1.5 wt.% decreases the hardness as shown by loading 2.5 wt.%. This observation can be referred to the reduction of the cohesive strength of the coated film [13]. It is also clear from the obtained results of the adhesion test that, the force required to pull-off the blank epoxy coating is smaller than those in the composite epoxy coating formulations (Table 2). It is observed a significant increase in the pull-off adhesion upon loading $ZrO_2/Cr_2O_3$ NPs to the epoxy coating. The improvement in adhesion property of epoxy coating is a positive result can be attributed to the reinforcement provided by $ZrO_2/Cr_2O_3$ NPs, their good dispersion in epoxy coating, and they can facilitate the curing of epoxide rings with polyamines [13].

The impact resistance of epoxy nanocomposite films (Table 2) was improved from 5 to 10 J at 1.5 wt.% $ZrO_2/Cr_2O_3$ NP epoxy composite coating. The decrease in impact resistance at higher loading more than 1.5 wt.% may be attributed to poor dispersion of $ZrO_2/Cr_2O_3$ NPs with the crosslinking reaction among epoxy matrix. The bend test was performed to study the flexibility of the blank epoxy coating and composite epoxy coating filled with $ZrO_2/Cr_2O_3$ NPs to confirm that there is no any

significant difference between a blank and composite epoxy coatings. The abrasion resistance of the blank and $ZrO_2/Cr_2O_3$ epoxy nanocomposites as organic coating films on the steel substrate was calculated as loss in weight at 1000 abrasion cycles. The obtained results of abrasion resistance (Table 2) show that the weight loss (mg) is gradually being reduced with an increase in the concentration of $ZrO_2/Cr_2O_3$ NPs from 0.5 to 1.5 wt.% from 65 to 25 mg relative to 85 mg for the blank epoxy. This improvement may be attributed to the enhancement interaction of $ZrO_2/Cr_2O_3$ NPs with epoxy structure. Finally, it can be conclude that the $ZrO_2/Cr_2O_3$ NPs produce more compact and less abraded epoxy nanocomposite coatings as compared to the blank epoxy due to increase the interface surface interaction between $ZrO_2/Cr_2O_3$ NPs and the epoxy matrix.

**Table 2.** Mechanical resistance of $ZrO_2/Cr_2O_3$ epoxy nanocomposite coating films at different at different $ZrO_2/Cr_2O_3$ NPs loading.

| Coating Design | NPs Weight % (wt.%) | Hardness (Newton) | Adhesion (MPa) | Impact (Joule) | Pending | Weight Loss (mg)/1000 Cycles |
|---|---|---|---|---|---|---|
| Blank epoxy | 0 | 5 ± 0.1 | 3 ± 1.8 | 5 ± 0.2 | Pass | 85 ± 2.2 |
| Epoxy/ZrO$_2$/Cr$_2$O$_3$ NPs | 0.5 | 10 ± 0.3 | 4 ± 0.2 | 7 ± 0.1 | Pass | 65 ± 1.4 |
| | 1.0 | 11 ± 0.2 | 6 ± 1.1 | 9 ± 0.3 | Pass | 40 ± 1.1 |
| | 1.5 | 12 ± 0.1 | 9 ± 1.4 | 10 ± 0.1 | Pass | 25 ± 2.1 |
| | 2.5 | 11 ± 0.2 | 7 ± 0.6 | 8 ± 0.2 | Pass | 30 ± 1.8 |

The corrosion resistance was studied to investigate the effect of $ZrO_2/Cr_2O_3$ NPs on the protective performances of the epoxy coatings. In this respect photographic reference standards were used to evaluate the degree of blistering and to determine the percentage of the area rusted after salt spray fog exposure as summarized in Table 3 and Figure 8a–e. The salt spray resistance data (Table 3 and Figure 8a–e) indicated that the corrosion resistance was significantly improved by the incorporation of $ZrO_2/Cr_2O_3$ NPs when compared to neat epoxy resin. This improvement which may be attributed to $ZrO_2/Cr_2O_3$ NPs are an inert lamellar filler, which orientate themselves parallel to the substrate surface and inhibiting corrosion by acting as a barrier element to water and oxygen from the environment [13,39].

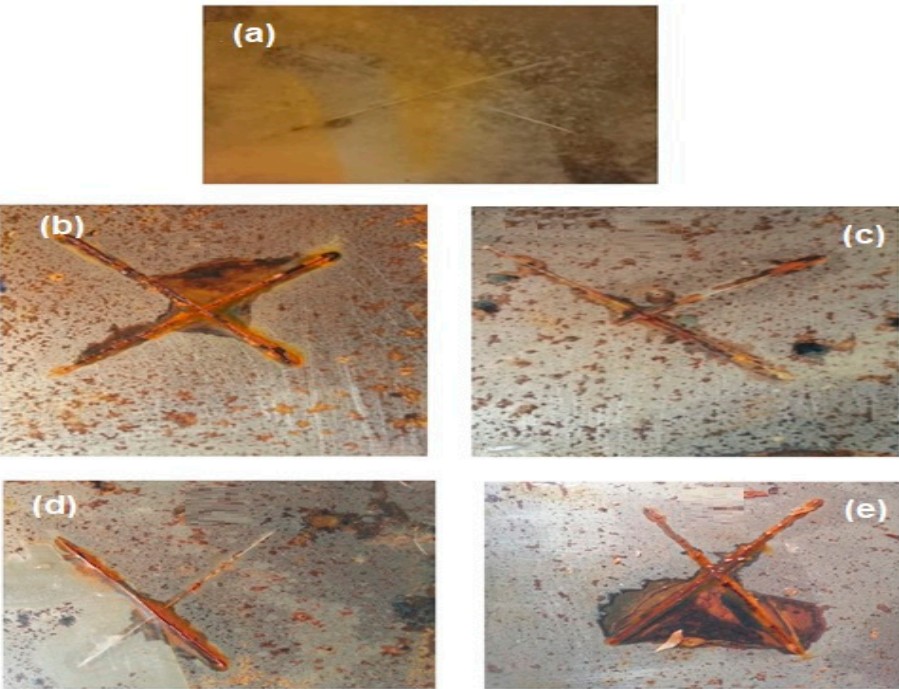

**Figure 8.** Salt spray resistance photos of epoxy coating mixed with different (**a**) 0, (**b**) 0.5, (**c**) 1.0, (**d**) 1.5, and (**e**) 2.5 wt.% of $ZrO_2/Cr_2O_3$ NPs coating films after 750 h of salt spray exposure.

**Table 3.** Salt spray resistance of $ZrO_2/Cr_2O_3$ epoxy nanocomposites coating films at different $ZrO_2/Cr_2O_3$ NPs loading after exposure time 720 h.

| Coating Design | NPs Weight % (wt.%) | Disbanded Area % | Rating Number (ASTM D-1654) |
|---|---|---|---|
| Blank epoxy | 0 | $19 \pm 0.1$ | 5 |
| Epoxy/$ZrO_2/Cr_2O_3$ NPs | 0.5 | $5 \pm 0.05$ | 7 |
| | 1.0 | $2 \pm 0.08$ | 8 |
| | 1.5 | $1 \pm 0.08$ | 9 |
| | 2.5 | $2 \pm 0.04$ | 8 |

## 4. Conclusions

$Cr_2O_3$ and $ZrO_2$ nanoparticles were synthesized separately and characterized to confirm that lower dispersion of $ZrO_2$ NPs than $Cr_2O_3$ NPs in water due to the formation of the hydration layer on the $Cr_2O_3$ NPs surfaces. The obtained results show that the increasing of $ZrO_2/Cr_2O_3$ NPs concentration from 0.5 to 2.5 wt.% increases the thermal stability of the epoxy to increase the initial degradation temperature of the epoxy networks from 235 °C to 285 °C at 10% weight loss. The nanocomposite containing 1.5 wt.% of $ZrO_2/Cr_2O_3$ NPs provided the best thermal stability, corrosion resistance, and mechanical properties such as adhesion, hardness, impact, and abrasion due to the formation $ZrO_2/Cr_2O_3$ NPs orientate themselves parallel to the substrate surface and inhibiting corrosion by acting as a barrier element to water and oxygen from the environment.

**Author Contributions:** Conceptualization, methodology, investigation, and writing—review and editing, A.M.A., M.A.A., and A.M.E.-S.; supervision, A.M.A., O.M.A.-E., and M.A.E.-S.; funding acquisition and data curation, A.M.A. and M.A.E.-S. All authors have read and agreed to the published version of the manuscript.

**Funding:** This project was financially supported by King Saud University with the researchers' supporting project number (RSP-2020/63), King Saud University, Riyadh, Saudi Arabia.

**Acknowledgments:** The authors acknowledge King Saud University, researchers supporting project number (RSP-2020/63), King Saud University, Riyadh, Saudi Arabia.

**Conflicts of Interest:** The authors declare no conflict of interest.

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
