# Peer review of "Hybrid ZrO2/Cr2O3 Epoxy Nanocomposites as Organic Coatings for Steel"

_coatings, doi:10.3390/coatings10100997_

Round 1

Reviewer 1 Report

The topic of the ZrO2 / Cr2O3 NPs on the properties of epoxy coating is interesting, but it could not be published in its present form.

1. The figure quality should be improved, such as Figure 6. There is 5 lines in it, but only three lables can be found.

2. The Figure 7 is a bit contradict to the conclusion that additive of the ZrO2 / Cr2O3 NPs improves the anti corrosion of the epoxy coating.

3. The adhesion between coating and steel is not a typical value of epoxy coating, maybe the preparation method is not correct.

Author Response

x

  1. The figure quality should be improved, such as Figure 6. There is 5 lines in it, but only three lables can be found.

Answer: All figures were clarified

  1. The Figure 7 is a bit contradict to the conclusion that additive of theZrO2 / Cr2O3 NPsimproves the anti corrosion of the epoxy coating.

Answer: Figure 7 show the salt spray photos after 750 h moreover the corrosion resistance was determined from the rust % under coatings that show as determined in Table 3/

  1. The adhesion between coating and steel is not a typical value of epoxy coating, maybe the preparation method is not correct.

Answer: The adhesion test of the epoxy on the steel surfaces is 5 MPa and the data listed in Table 2 determined from the experimental works at the same condition the standered error values added.

Reviewer 2 Report

The manuscript is focusing on the preparation and characterization of epoxy composite films modified with ZrO2 and Cr2O3 nanoparticles for improved mechanical and thermal characteristics of the coatings. The paper contains original results and deserves to be published after minor revision. My comments are as follows:

  1. Page 3, line 110: How could the authors be sure that by spraying, the film thickness was exactly 100 microns? I also suggest replacing the abbreviation DFT (which is most commonly used for “Density Functional Theory” with “DF’ or “UDFT”).
  2. Page 4, line 144: Please replace “performed” with ”investigated”.
  3. Page 4, line 149: I suggest eliminating “study” from the title of section 3.1. Same advice for the title of section 2.3. It is redundant.
  4. Figure 4. In order to compare the different samples, it is recommended that the scale of the micrographs is the same.
  5. Legend of Figure 5: The different loading levels should be expressed in %.
  6. Table 1 can be rearranged: As all the samples are Epoxy / ZrO2 / Cr2O3 NPs, the label can be written only once, in a separate column, and the loadings in another column. Same recommendation for Tables 2 and 3.
  7. Table 3: As all the experiments lasted 720 h, it is not necessary to introduce this value in the table. It is enough to mention it in the title of the table.
  8. Legend of Figure 7 is not clear.

Author Response

 My comments are as follows:

  1. Page 3, line 110: How could the authors be sure that by spraying, the film thickness was exactly 100 microns? I also suggest replacing the abbreviation DFT (which is most commonly used for “Density Functional Theory” with “DF’ or “UDFT”).

Answer: it is corrected to DT

  1. Page 4, line 144: Please replace “performed” with ”investigated”.

Answer: It is corrected

  1. Page 4, line 149: I suggest eliminating “study” from the title of section 3.1. Same advice for the title of section 2.3. It is redundant.

Answer: detleted

  1. Figure 4. In order to compare the different samples, it is recommended that the scale of the micrographs is the same.

Answer: New SEM added and all changed to be 30 µm except c that only available

  1. Legend of Figure 5: The different loading levels should be expressed in %.

Answer: Figure corrected

  1. Table 1 can be rearranged: As all the samples are Epoxy / ZrO2 / Cr2O3 NPs, the label can be written only once, in a separate column, and the loadings in another column. Same recommendation for Tables 2 and 3.

Answer: Corrected

  1. Table 3: As all the experiments lasted 720 h, it is not necessary to introduce this value in the table. It is enough to mention it in the title of the table.

Answer: Table modified

  1. Legend of Figure 7 is not clear.

Answer: Figure 7 clarified

Round 2

Reviewer 1 Report

The manuscrip had been improved significantly and can be accepted for publication.